# Exploring experiences of times without care and encounters in dementia: protocol for a living and adaptive evidence map

Julian Hirt ©,[1,2] Laura Adlbrecht ©,[1] Carola Maurer ©,[1] Thomas Beer[1]

¹Department of Health, Eastern Switzerland University of Applied Sciences, St.Gallen, Switzerland
²Pragmatic Evidence Lab, Research Center for Clinical Neuroimmunology and Neuroscience Basel (RC2NB), University of Basel and University Hospital Basel, Basel, Switzerland

**Correspondence to**
Julian Hirt; julian.hirt@ost.ch

## ABSTRACT

**Introduction** Individuals with dementia spend most of the day without care, without encounters, and usually without activity. Although this has been proven in studies, there is a knowledge gap on how individuals with dementia experience these periods of time. Such knowledge would be highly relevant for health professionals and relatives to develop adequate strategies for dealing with these periods of time. The *FreiZeit* study aims to reconstruct periods of time without care and encounters from the perspective of individuals with dementia and formal and informal carers. The specific objective of this review is to provide a continuously updated overview of the topical evidence that may be used to guide data synthesis and interpretation within the *FreiZeit* study.

**Methods and analysis** We conduct a living evidence map, based on a comprehensive systematic literature search in MEDLINE/PubMed, CINAHL, PsycINFO/Ovid and Web of Science Core Collection, citation-based searches and web searches. We include studies on times without care and encounters of individuals with dementia from the perspective of individuals with dementia themselves and formal or informal caregivers of any observational study design that were conducted in the institutional and domestic long-term care setting and published as journal article in English, French or German language without any restriction of the publication year. One reviewer screens titles, abstracts and full texts and extracts data. Key characteristics and results of the included studies are charted in a tabular format. The searches will be run and continuously updated throughout the duration of the overarching *FreiZeit* study (every 6 months for 2 years from 2023 to 2025).

**Ethics and dissemination** Ethics approval is not required for this evidence map. We disseminate our findings via journal articles and conference proceedings as well as other formats.

**Registration details** This review protocol is uploaded on Open Science Framework (OSF; DOI 10.17605/OSF.IO/GDYZ9).

## STRENGTHS AND LIMITATIONS OF THIS STUDY

⇒ A comprehensive literature search combining electronic key database searching, citation-based searching and web searching with no restrictions on publication date and outcomes is a methodological strength of our review.
⇒ A living approach enables the continuous updating and monitoring of newly published studies.
⇒ By an adaptive approach, we aim at enabling to modify the search strategy based on newly retrieved eligible studies or research questions that may be complemented.

majority of the population feels well informed about dementia and believes that there are good options for supporting individuals with dementia.[1] At the same time, however, there is a marginalisation of dementia at various levels. A review by Lara *et al* shows the connection between subjectively perceived loneliness and the risk of developing dementia.[2] Accordingly, individuals with a subjective sense of marginalisation are particularly affected by dementia. They feel socially excluded even before diagnosis. On the other hand, the pathologising or stigmatising individuals with dementia as 'dementia patients' leads to an increase in marginalisation. With, despite, and because of their varying degrees of loss of cognitive and communicative abilities, they are less and less integrated into social life in the sense of social participation.

The care for individuals with dementia care is increasingly provided in a social protective environment,[3] for example, in a family-based, dyadic care dependency or in a care facility. In Switzerland, one out of four nursing homes care for individuals with dementia in so-called protected living areas, which are mainly closed or technologies are used to surveil individuals who left the area.[4] As a result, individuals with dementia become location-fixated, immobile and, thus, isolated.[5 6] The

## INTRODUCTION

For several years, the topic of dementia has gained an enormous position in the political, social and scientific debate. As the Swiss Dementia Barometer 2018 shows, the

**BMJ** 1

lack of adequate cognitive, emotional and sensory stimulation has a negative impact on the course of dementia, the development of challenging behaviours, the shaping of the relationship work with their environment, and, above all, on the physical, cognitive, and emotional state.[7]

There is ample empirical evidence that individuals with dementia in institutional care settings experience long periods of boredom, inactivity, sleep and associated periods of loneliness,[8–11] indicating that individuals with dementia experience large periods of time in their daily routine when they are left to their own, that is, without any encounters where they are directly addressed by another person such as caregivers, other residents, family members. However, it lacks a systematic evidence overview on the characterisation, temporal and periodical extent and how individuals with dementia and caregivers experience times without care and encounters or times in which (nearly) nothing obviously happens to and/or with individuals with dementia. Such knowledge would be highly relevant for nursing homes, health professionals and individuals' relatives to develop adequate strategies for detecting, assessing and dealing with these periods of time.

Our primary objective is to search and map the topical evidence on times without care and encounters of people with dementia; explicitly guided by the following research questions (RQ):

- RQ1: What happens in times without care and encounters of individuals with dementia?
- RQ2: To what temporal and periodical extent do individuals with dementia experience times without care and encounter?
- RQ3: How do individuals with dementia experience times without care and encounters from their own or foreign perspective (such as caregivers and researchers)?
- RQ4: How do formal and informal caregivers experience times without care and encounters of individuals with dementia?

Our secondary objective with this review is to provide evidence for the conduct of an empirically ethnography that aims to reconstruct periods of time without care and encounters (referred to as leisure time, in German *Freizeit*) from the perspective of the individuals with dementia and from the point of view of formal or informal carers in the institutional and domestic long-term care setting in Switzerland and Germany.[12–14] Evidence that is collected by our review may support and guide the data collection, synthesis and interpretation within the *FreiZeit* study (separate protocol in preparation[12]).

## METHODS AND ANALYSIS
### Design
We generate a living and adaptive evidence map, based on a comprehensive systematic literature search.[15] An evidence map aims at mapping out, categorising and charting existing topical literature without its formal critical or quality assessment or synthesis.[16] A living approach enables the continuous monitoring, consideration and integration of newly published studies into the body of evidence throughout the duration of the *FreiZeit* project.[12–14] By an adaptive approach, we aim at enabling to modify the search strategy based on newly retrieved eligible studies or research questions that may be complemented. A detailed methodological outline is provided below, structured following the Preferred Reporting Items for Systematic Reviews and Meta-Analyses guidance.[17] This review protocol is registered on Open Science Framework (OSF[18]).

### Eligibility criteria
We include studies on times without care and encounters of individuals with dementia from the perspective of individuals with dementia themselves and formal or informal caregivers of any quantitative observational and qualitative study design (ie, cohort study, cross-sectional study, survey, qualitative descriptive study, ethnography), which were conducted in the institutional and domestic long-term care setting and published as journal articles in English, French or German language without any restriction of the publication year (table 1); with no restriction to sample size and type and severity of dementia. Eligible studies refer to a sample of individuals with dementia of at least 75% of the overall study sample. A formal diagnosis of dementia is no prerequisite for individuals to be eligible for our review.

### Information sources
We search MEDLINE via PubMed, CINAHL, PsycINFO via Ovid and the Web of Science Core Collection.[19] In addition, we conduct web searches using Google Scholar and citation searching using Scopus (ie, backward and forward citation searching of eligible publications and pertinent evidence syntheses that are retrieved during study selection until no further eligible publications can be identified)[20] and Local Citation Network (ie, incoming and outgoing suggestions).[21]

### Search strategy
The search strategy is based on the concepts dementia (population), times without care and encounters (phenomenon of interest) and the study setting (institutional and domestic long-term care setting; table 2).

The collection of search terms (table 2) is based on an initial search and meta-data of its results (ie, title, abstract, author keyword(s), controlled vocabulary),[22] brainstorming for initial search terms with members of the *Frei-Zeit* study team (LA and CM), reviews that were previously worked on by members of the review team and the experience in designing dementia-specific search strings of the review team[23–28] and the results of ongoing preliminary searches (meta-data, ie, title, abstract, author keyword(s), controlled vocabulary). The controlled vocabulary search is conducted within the specific database-controlled vocabulary catalogues and is supported by the use of

**Table 1** Eligibility criteria following the population-interest-context (PICo) format

| PICo | Inclusion criteria | Exclusion criteria |
|---|---|---|
| **P**opulation | Individuals with dementia * <br> Formal caregivers † <br> Informal caregivers ‡ | Any other |
| Phenomenon of **I**nterest | Times without care and encounters | Other times |
| **C**ontext | Institutional long-term care (eg, nursing home, care home, assisted living, day care) <br> Domestic long-term care (ie, home care) | Acute care <br> Primary care |
| Design | Any type of quantitative observational or qualitative study | Evidence syntheses <br> Experimental/interventional research |
| Publication type | Journal article § | Any other |
| Publication year | Any | – |
| Language | English <br> French <br> German | Any other |

*As referred to by study authors that may mean that dementia is formally diagnosed by a physician and or assessed by health professionals such as nurses or psychologists.
†Professional caregivers such as nurses, nurse aids or activity staff.
‡Non-paid caregivers such as family members, relatives or friends.
§Irrespective of whether an article was peer-reviewed or not.

the Yale MeSH Analyzer,[29] PubMed PubReminer[30] and Coremine Medical[31] based on references retrieved by the initial and preliminary searches.[22]

A PubMed search of MEDLINE is translated to other databases using the Polyglot Search tool[32] and manual confirmation or adaptions by one reviewer (JH). The database-specific search strategies (for the initial search in May 2023) are provided in the online supplemental appendix 1 (any updates on the database-specific search strategies will be reported on OSF).[18]

For web searching via Google Scholar, we use a phrase derived from the research questions and title of our review (ie, 'dementia times without care and encounters', considering the first 20 hits). For citation searching, we perform backward citation searching (reference list checking) and forward citation searching (cited-by search) using all eligible studies and pertinent evidence syntheses (that are retrieved during study selection) as seed references via Scopus[33]; we perform iterative layers until no more additional relevant reference retrieves. In addition, we use the function of incoming and outgoing suggestions of the Local Citation Network to retrieve references that were not yet identified via other sources.[21]

All searches are carried out approximately every 6 months for the duration of the *FreiZeit* project (each May and November of the years 2023 and 2024).

### Study selection and data extraction
One reviewer (JH) screens titles, abstracts and full texts for eligibility using the Rayyan web app[34] (title/abstract level by screening all references manually, ie, without the use of integrated machine learning features) and Covidence[35] (full-text level). Study selection is confirmed by a second reviewer (TB), if necessary.

One reviewer (JH) extracts the following data based on the electronic full-text article version(s) using Covidence[35]: author(s) and publication year, country of conduct, study aim or research question (as expressed by authors) and which RQs are addressed, study design, population (type of participants, sample size and type and severity of dementia (as per eligibility criteria), setting (type as expressed by authors), number of study sites, data collection (methods (eg, observation, survey), assessment instrument, perspective, and, for observations, period of time (eg, x days between month y and z), time frame (eg, between x and y o'clock) and schedule (eg, any other details provided)), data analysis methods, unit of observation (ie, observational assessment or individuals with dementia) and main results. If the reported results of individual studies are not adequately usable for the purpose of our evidence map (ie, depending on the way the data are presented), we track this accordingly, but we do not request data from study authors. Data extraction is confirmed by a second reviewer (TB), if necessary. The list of variables that are extracted is not exhaustive and may be expanded, if deemed necessary by the review and/or the *FreiZeit* team.

### Critical appraisal
As this is an evidence map that aims at providing an overview of the available topical evidence without any quality assessment, we do not critically appraise the included studies.

### Data charting
Key characteristics and results of the included studies are charted in a tabular format with columns that are

**Table 2** Collection of search terms (as of when submitting this manuscript; without any search techniques added (eg, wildcards), without indicating controlled vocabulary)

| Search component | Search terms |
|---|---|
| Dementia | ALZHEIMERS DISEASE<br>DEMENTIA |
| Times without care and encounters | ASSESSMENT TOOL FOR OCCUPATION AND SOCIAL ENGAGEMENT<br>ATOSE<br>BORED<br>BOREDOM<br>BORING<br>CARE TIME<br>DCM<br>DEMENTIA CARE MAPPING<br>DEMENTIA-CARE MAPPING<br>DISCONNECTEDNESS<br>CONNECTEDNESS<br>CONTACTLESS<br>DETACHED<br>EVERYDAY LIFE<br>EXPERIENCE OF TIME<br>FEELING OF TIME<br>FREE TIME<br>INACTIVE<br>INACTIVITY<br>LEISURE<br>LIFEWORLD<br>LONESOME<br>MAASTRICHT ELECTRONIC DAILY LIFE OBSERVATION TOOL<br>MEDLO-TOOL<br>MENORAH PARK ENGAGEMENT SCALE<br>MPES<br>MONOTONY<br>NO CARE<br>NO CONTACT<br>NO SOCIAL INTERACTION<br>NOT ENGAGED<br>NOT INVOLVED<br>RESPITE<br>RESTRAINED<br>RETREATED<br>SENSE OF TIME<br>SOCIABILITY<br>TEMPORAL PERCEPTION<br>TIME PERCEPTION<br>UNENGAGED<br>UNOCCUPIED<br>UNSTRUCTURED<br>WITHDRAWAL<br>WITHDRAWN |

Continued

**Table 2** Continued

| Search component | Search terms |
|---|---|
| Setting (institutional and domestic long-term care) | *(institutional long-term care)*<br>ASSISTED LIVING FACILITY<br>CARE CENTRE<br>CARE CENTER<br>CARE HOME<br>DAY CARE<br>DAY-CARE<br>DAY HOSPITAL<br>DAY-HOSPITAL<br>HOME FOR AGED<br>HOME FOR THE AGED<br>HOUSING FOR THE ELDERLY<br>LONG TERM CARE<br>LONGTERM CARE<br>LONG-TERM CARE<br>LONG-TERM RESIDENT<br>LONG-TERM RESIDENTIAL<br>LTC<br>NURSING HOME<br>OLD AGE HOME<br>OLD PEOPLE'S HOME<br>RESIDENT<br>RESIDENTIAL AGED CARE FACILITY<br>RESIDENTIAL FACILITY<br>RESIDENTIAL SETTING<br>REST HOME<br>RETIREMENT HOME<br>SENIOR-CITIZENS HOME<br>SENIOR CITIZENS HOME<br>*(domestic long-term care)*<br>AMBULATORY CARE<br>COMMUNITY-DWELLING<br>DOMESTIC<br>DOMICILIARY CARE<br>HOME<br>HOME BASED<br>HOME CARE<br>HOME HEALTH<br>HOME HEALTH CARE<br>HOME HEALTH NURSING<br>HOME NURSING<br>HOME-BASED<br>HOME-CARE<br>LIVING IN THE COMMUNITY<br>NON-INSTITUTIONAL CARE<br>NON-INSTITUTIONALIZED CARE |

based on the extracted variables (see chapter study selection and data extraction).

## Data management

One reviewer (JH) is responsible for data and literature management. A central Citavi literature database contains all search hits.[36] Newly identified search results, irrespective of data source, are deduplicated against the existing literature in the database using the Citavi deduplication function.[36]

## ETHICS AND DISSEMINATION

For each update of the evidence, we provide the applied methods and a tabular overview of the studies and their results on OSF.[18] The final version of the evidence will be

aimed to publish in an open access international peer-reviewed journal.

In addition, we aim at disseminating our findings in scientific and non-scientific journal articles and conference proceedings as well as formats directed to the public and decision-makers in healthcare.

**Contributors** JH and TB had the idea for this review and made substantial contributions to its conception. JH, LA, CM and TB designed the methodological outline for this review. JH drafted the manuscript. LA, CM and TB critically revised it for important intellectual content. JH, LA, CM and TB (all authors) finally approved the latest version to be published and agreed to be accountable for all aspects of the work in ensuring that questions related to the accuracy and integrity of any part of the work are appropriately investigated and resolved.

**Funding** Swiss National Science Foundation (Weave/Lead Agency; project number: 200919). The funder had no influence on the design, conduct, and reporting of this review.

**Competing interests** None declared.

**Patient and public involvement** Patients and/or the public were not involved in the design, or conduct, or reporting, or dissemination plans of this research.

**Patient consent for publication** Not applicable.

**Provenance and peer review** Not commissioned; externally peer reviewed.

**Data availability statement** All data used for the design of this review is published alongside.

**ORCID iDs**
Julian Hirt http://orcid.org/0000-0001-6589-3936
Laura Adlbrecht http://orcid.org/0000-0002-7042-7523
Carola Maurer http://orcid.org/0000-0002-2731-7767

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
