## [Reviewer comments · BMJ Open]

ARTICLE DETAILS

TITLE (PROVISIONAL)	Exploring experiences of times without care and encounters in dementia: protocol for a living and adaptive evidence map
AUTHORS	Hirt, Julian; Adlbrecht, Laura; Maurer, Carola; Beer, Thomas

VERSION 1 – REVIEW

REVIEWER	Ward, Richard University of Stirling
REVIEW RETURNED	19-Jun-2023

GENERAL COMMENTS	This is a really interesting protocol and I look forward to reading the results of this rolling review. I would agree with your anticipated limitation that the available evidence may be limited and I wonder if there are any steps you can take to complement the proposed review with data or insights from other sources? For instance, I know that the care home inspection process (in the UK) utilises SOFI (short observational framework for inspection) that might yield some insights and other dementia care mapping reports could also be worth a look. In table 2 I noticed that you hadn't included the term 'occupation' which would be a good idea. The accompanying literature on occupational justice (from an Occupational Therapy perspective on dementia) might also be worthwhile looking at. My only other query concerns your perspective on time itself, given that we know that the experience and perception of time by people with dementia can be altered. There seems to be an assumption in the protocol that our understanding of time is limited to a chronological and linear conception and I wonder if it might be helpful to consider further or alternative understanding? Overall, the study promises to deliver some important insights and useful knowledge that could inform care planning and practice.
---

REVIEWER	Roes, Martina German Center for Neurodegenerative Diseases
REVIEW RETURNED	17-Jul-2023

GENERAL COMMENTS	please clarify who the population is (people with dementia: does this mean they will be diagnosed by a physician and/or will people with symptoms related to dementia be included too? (see table 1) its not always clear who is meant by 'caregiver/carer' (professionals vs family vs other relevant people of the social network) (see table 1) line 34/35 (p4 - introduction): from my understanding there are conceptual differences within the mentioned 'observations of ...'
---

	e.g. sitting alone". Just sitting alone does not automatically mean that the person experiences feelings of loneliness; observations of 'resting, dozing, sleeping' need to be differentiated from 'being inactive' ... sleeping during the day can be caused by different reasons, bringing it up here (line 37) seems to be decontextualized and presented in a negative way line 42/43 (p4 - introduction) : please make sure that the reader understands what is meant by 'passive, non-purposeful activity or inactive' ...specifically with regard to the mentioned observations (line 35/56, P4) line 48/49 (p4)- please clarify what is meant by 'encounters' (see also table 1) line 57 (p4) - research question 1: does this include 24/7 or will hours of interest be defined? line 58 (p4) - research question 2: will there be done a comparison of living arrangements? line 1 (p5) - not clear what is the difference to research Q1? line 2 (p5) same as in line 1 line 5/6 (p5) - see comment above: who is meant by informal/formal caregiver line 9/10 (p5) - how will Freizeit be defined across all care settings? line 13/14 (p5) - the other protocol would be needed to understand this paper. please add general definition here too line 35 (p5) - will only peer-reviewed article be included or grey literature too? (see also table 1) line 32/33 (p.6) - the argument presented here is not sufficient, evidence would indicate critical appraisal, thus provide other/additional reasons line 24/25 (p6) - please add a critical reflection of using AI (Ryvan App) for the review (maybe under ethics - research ethics??) table 2 (p.8) : why are 'care time', 'dementia care mapping' and 'restraints' listed as a search term under 'times without care and encounters'? table 2 (p.8): why is 'day hospital' listed, table 1 say acute hospital are excluded
--	---

VERSION 1 – AUTHOR RESPONSE

Reviewer: 1

Dr. Richard Ward, University of Stirling

Comments to the Author:

This is a really interesting protocol and I look forward to reading the results of this rolling review. I would agree with your anticipated limitation that the available evidence may be limited and I wonder if there are any steps you can take to complement the proposed review with data or insights from other sources? For instance, I know that the care home inspection process (in the UK) utilises SOFI (short observational framework for inspection) that might yield some insights and other dementia care mapping reports could also be worth a look. In table 2 I noticed that you hadn't included the term 'occupation' which would be a good idea. The accompanying literature on occupational justice (from an Occupational Therapy perspective on dementia) might also be worthwhile looking at. My only other

query concerns your perspective on time itself, given that we know that the experience and perception of time by people with dementia can be altered. There seems to be an assumption in the protocol that our understanding of time is limited to a chronological and linear conception and I wonder if it might be helpful to consider further or alternative understanding? Overall, the study promises to deliver some important insights and useful knowledge that could inform care planning and practice.

We agree that the understanding of time of persons with dementia does not seem to be exclusively oriented towards a chronological or linear idea. Their understanding of time does not correlate - and if so, only weakly - with that of persons without dementia (Honer, 2011). Therefore, we are especially interested in the temporal experience of persons with dementia of times without care and encounters and address this in depth in our ethnographic study.

We dropped the limitation on the number of assumed eligible studies since we already identified around 30 studies during the first round of our review search. This makes us confident that we will face more published studies as preliminary expected.

We considered the term occupation when designing our search strategies but realized that it retrieved too many references that were not relevant, i.e., when aiming to search specifically for times without occupation. Hence, we would like to avoid taking up this term again. Nevertheless, we identified relevant literature that contextualizes occupational justice as suggested by you.

We agree that the understanding of time of persons with dementia does not seem to be exclusively oriented towards a chronological or linear idea. Their understanding of time does not correlate - and if so, only weakly - with that of persons without dementia (Honer, 2011). Therefore, we are especially interested in the temporal experience of persons with dementia of times without care and encounters and address these in depth in our ethnographic study.

Reviewer: 2

Prof. Martina Roes, German Center for Neurodegenerative Diseases

Comments to the Author:

please clarify who the population is (people with dementia: does this mean they will be diagnosed by a physician and/or will people with symptoms related to dementia be included too? (see table 1)

>>>> We added information in the section eligibility criteria and table 1 and consider dementia "as referred to by study authors that may mean that dementia is formally diagnosed by a physician and or assessed by health professionals such as nurses or psychologists".

its not always clear who is meant by 'caregiver/carer' (professionals vs family vs other relevant people of the social network) (see table 1)

>>>> We elaborated on this in footnote of table 1: "formal caregivers are considered as professional caregivers such as nurses, nurse aids, or activity staff; informal caregivers are considered as non-paid caregivers such as family members, relatives, or friends".

line 34/35 (p4 - introduction): from my understanding there are conceptual differences within the mentioned 'observations of ...'

e.g. sitting alone". Just sitting alone does not automatically mean that the person experiences feelings of loneliness;

observations of 'resting, dozing, sleeping' need to be differentiated from 'being inactive' ...

sleeping during the day can be caused by different reasons, bringing it up here (line 37) seems to be decontextualized and presented in a negative way

>>>> We dropped this paragraph because we already referred to eligible studies in it.

line 42/43 (p4 - introduction) : please make sure that the reader understands what is meant by 'passive, non-purposeful activity or inactive' ...specifically with regard to the mentioned observations (line 35/56, P4)

>>>> See above, we dropped this paragraph.

line 48/49 (p4)- please clarify what is meant by 'encounters' (see also table 1)

>>>> We now specified as follows "indicating that individuals with dementia experience large periods of time in their daily routine when they are left to their own, i.e., without any encounters where they are directly addressed by another person such as caregivers, other residents, family members".

line 57 (p4) - research question 1: does this include 24/7 or will hours of interest be defined?

>>>> We assume day hours that were typically assessed; however, we would like to refrain from specifying this and to watch out which time periods are assessed in eligible studies.

line 58 (p4) - research question 2: will there be done a comparison of living arrangements?

>>>> As we extract the setting in which a study took place, we will report this in our results.

line 1 (p5) - not clear what is the difference to research Q1?

line 2 (p5) same as in line 1

>>>> We condensed these two research questions with research question 1.

line 5/6 (p5) - see comment above: who is meant by informal/formal caregiver

>>>> We revised as described above.

line 9/10 (p5) - how will Freizeit be defined across all care settings?

>>>> FreiZeit is the short form of the project title (leisure or free time) and should be seen as a pun with the overall project goal that is to explore periods of time without care and encounters of people with dementia.

line 13/14 (p5) - the other protocol would be needed to understand this paper. please add general definition here too

>>>> The overall study protocol is not available yet but we added a reference to a project outline: <https://osf.io/uxt48>

line 35 (p5) - will only peer-reviewed article be included or grey literature too? (see also table 1)

>>>> We will consider journal articles with no restriction whether an article was peer-reviewed or not. We added this detail to the footnote.

line 32/33 (p.6) - the argument presented here is not sufficient, evidence would indicate critical appraisal, thus provide other/additional reasons

>>>> We revised accordingly: "As this is an evidence map that aims at providing an overview of the available topical evidence without any quality assessment, we do not critically appraise the included

studies".

line 24/25 (p6) - please add a critical reflection of using AI (Rayyan App) for the review (maybe under ethics - research ethics??)

>>>> As we use Rayyan as a tool and screen every reference manually, we are not using any machine learning feature and specified accordingly: "... screens titles, abstracts, and full texts for eligibility using the Rayyan web app [42] (title/abstract level by screening all references manually, i.e., without the use of integrated machine learning features) and...".

table 2 (p.8) : why are 'care time', 'dementia care mapping' and 'restraints' listed as a search term under 'times without care and encounters'?

>>>> We consider these terms, and also assessment instruments such as DCM or the MEDLO-tool, as highly important to search for our phenomena of interest (i.e., times without care and encounters).

table 2 (p.8): why is 'day hospital' listed, table 1 say acute hospital are excluded

>>>> We are aware that day hospital may be referred to as an outpatient facility where patients attend for assessment, treatment or rehabilitation during the day and then return home or spend the night at a different facility; as a type of day care/rehabilitation, we would consider it for eligibility.